# Turning the Tide—Artificial Intelligence in the Evolving Landscape of Liver Cancer

**DOI:** 10.3390/cancers17183003

**Published:** 2025-09-14

**Authors:** Cristiana Grapă, Tudor Mocan, Lavinia Patricia Mocan, Andrei Motofelea, Raluca Stănciulescu, Rareș Crăciun, Andrei Vârciu, Zeno Spârchez, Teodora Mocan

**Affiliations:** 1Gastoenterology Clinic, “Prof. Dr. O. Fodor” Regional Institute of Gastroenterology and Hepatology, 400162 Cluj-Napoca, Romania; grapa.cristiana.maria@elearn.umfcluj.ro (C.G.); ioanmocan1@ubbcluj.ro (T.M.); ionel.andr.motofelea@elearn.umfcluj.ro (A.M.); stanciulescu_raluca_maria@elearn.umfcluj.ro (R.S.); craciun.rares.calin@elearn.umfcluj.ro (R.C.); zsparchez@elearn.umfcluj.ro (Z.S.); 2Department of Physiology, “Iuliu Hatieganu” University of Medicine and Pharmacy, 400012 Cluj-Napoca, Romania; teodora.mocan@elearn.umfcluj.ro; 3UBBmed Department, Babes,-Bolyai University, 400084 Cluj-Napoca, Romania; 4Department of Histology, Iuliu Haţieganu University of Medicine and Pharmacy, 400012 Cluj-Napoca, Romania; 5Department of Internal Medicine, “Iuliu Hatieganu” University of Medicine and Pharmacy, 400012 Cluj-Napoca, Romania; 6Department of Internal Medicine, Emergency County Clinical Hospital Bistrita, 420094 Bistrita, Romania; contact@sjub.ro; 7Nanomedicine Department, Regional Institute of Gastroenterology and Hepatology, 400162 Cluj-Napoca, Romania

**Keywords:** artificial intelligence, liver cancer, diagnosis, treatment, precision medicine

## Abstract

Liver cancer is a major global health challenge and a leading cause of cancer-related deaths, with only modest progress in patient survival over recent years. This review examines the promise of artificial intelligence (AI) in transforming the way liver cancer is diagnosed, staged, treated, and monitored for recurrence. AI can combine information from medical imaging, tissue analysis, molecular testing, and patient records to identify patterns invisible to clinicians, supporting more tailored and accurate decisions. While technology holds great potential, many AI tools are still in development and face obstacles such as limited clinical testing, regulatory restrictions, and data protection issues. Our aim is to showcase how AI could be applied across all stages of liver cancer care, highlight the current gaps that must be addressed, and encourage collaboration between medical and technical experts to bring these innovations into everyday clinical practice.

## 1. Introduction

The liver, the body’s largest organ, plays a pivotal role in metabolism and detoxification. Liver tumors, with distinct types differing in cause, treatment, and prognosis, pose the same necessity for a holistic approach to properly diagnose and manage them [1]. Liver cancer is the third leading cause of cancer-related deaths worldwide [2], and can be broadly classified into hepatocellular carcinoma (HCC), cholangiocarcinoma (CCA)—which further comprises intrahepatic CCA (iCCA), perihilar CCA (pCCA) and distal CCA (dCCA), and liver metastases (LM)—the most common one being colorectal cancer liver metastases (CRLM) [3,4].

Hepatocellular carcinoma represents 90% of primary liver cancers [5] and creates significant challenges to both the patient and physician. Liver imaging is essential for tumor diagnosis and staging. Non-invasive diagnosis should be based on the LIRADS CT/MR v2018 [6]; conversely, if imaging criteria are not met, tumor biopsy is mandatory [5]. Once diagnosed, its management plan differs depending on the tumor stage, underlying chronic disease, and patient functional status; multiple treatment strategies are available, such as surgical resection (SR), liver ablation (LA), liver transplantation (LT), for early-stage HCC, while transarterial chemoembolization and systemic therapy are reserved for advanced HCC [5]. The Barcelona Clinic Liver Cancer prognosis and treatment strategy for HCC was last updated in 2022 [7] and summarizes the best clinical practice advice regarding HCC staging and management strategies. CCA comprises about 10–15% of primary liver cancers, arising from the biliary epithelium, peribiliary glands, or hepatic progenitor cells; its mortality varies depending on the type and stage, most patients being diagnosed in late stages, with a median dismal survival rate of approximately 24 months following diagnosis [8]. Diagnosis is mainly based on imaging modalities, with magnetic resonance cholangiopancreatography having the highest specificity and sensitivity for its detection [9]. For early-stage disease, the only curative treatment is surgical resection [8].

To alleviate the growing burden of liver cancer, innovative and comprehensive strategies are essential. These approaches must address a range of needs—from prevention and early detection to non-invasive diagnosis, prognosis, effective therapies, and evaluation of treatment response. During the coronavirus disease pandemic, the landscape of AI has expanded significantly from a theoretical construct, in a global effort to alleviate the saturation of the healthcare system. Artificial intelligence holds the potential to revolutionize care, enabling advancements in early identification and the development of personalized treatment plans [10,11].

AI is broadly defined as the use of computer programs to carry out complex tasks that mirror human cognitive functions, such as learning, reasoning under uncertainty, problem-solving, and representing knowledge. In clinical practice, AI encompasses a myriad of different technologies such as machine learning (ML), deep learning (DL), and natural language processing (NLP) (Figure 1) [12].

ML refers to algorithms that learn from data to perform tasks without being explicitly programmed [13]. It includes supervised learning, where models are trained on labeled data to predict outcomes, and unsupervised learning, which identifies patterns in unlabeled data. Reinforcement learning enables models to improve through feedback from their actions. In healthcare, ML is commonly used to classify patients and predict clinical outcomes, with supervised learning being the most widely applied approach [14].

DL, a branch of ML, uses neural networks modeled after the human brain to process complex, high-dimensional data [15]. Deep neural networks (DNNs) consist of multiple layers of interconnected neurons that transform input data through mathematical operations to generate predictive outputs [16]. Convolutional neural networks (CNNs), commonly used in medical imaging, detect spatial features and are effective in identifying liver lesions in CT or MRI scans. Popular CNN models include AlexNet, VGGNet, ResNet, YOLO, and U-Net [17]. Recurrent neural networks (RNNs), on the other hand, are designed for sequential data, enabling applications like text analysis and temporal data prediction [18].

A significant portion of patient information is recorded in unstructured formats, such as narrative clinical notes. However, this type of data is not easily analyzed using traditional machine learning methods. Natural language processing (NLP) addresses this challenge by automating the extraction of meaningful clinical features from text. These extracted features can be utilized for various purposes, including clinical surveillance and as input variables for other predictive modeling techniques [19].

In this narrative review, we focus on the potential of AI use for diagnosis, prognosis, treatment, and follow-up of the liver cancer patient, concentrating mainly on HCC, and including other types of liver cancers where evidence is available. We address the main state of AI, its potential, and the barriers that need to be overcome, with the aim of improving digital literacy among healthcare providers and enabling a superior standard of care for the liver cancer patient.

## 2. Methods

The current study is a narrative review that encompasses most relevant and comprehensive studies on artificial intelligence applications in liver cancer. Our research was based on the following search terms used in different combinations: “artificial intelligence”, “liver cancer”, “hepatocellular carcinoma”, “machine learning”, “deep learning”.

The inclusion criteria are represented by the following items: (1) original peer-reviewed studies, narrative and systematic reviews that cover the potential use of AI for diagnosis, staging, prognosis, treatment planning, or recurrence detection; (2) English language publications; (3) studies that include performance metrics such as area under the curve (AUC), specificity, sensitivity, concordance index, and validation strategy. Exclusion criteria: (1) non-English publications; (2) other types of studies such as case reports, editorials, letters, or conference papers. For each study, the following data were recorded: performance metrics of the AI method used, validation strategy (internal, cross-validation, external cohort—highlighting the articles that have external or prospective validation), and comparisons with standard-of-care (SOC) strategies, where applicable.

The results of the study are categorized according to the clinical pathway of the liver cancer patient: early detection, diagnosis, prognosis, treatment planning, and recurrence detection.

## 3. AI and the Journey of a Liver Cancer Patient

Research objectives for AI applications in liver cancer detection and management are usually designed to address specific unmet needs and overcome current limitations, such as operator experience dependence in ultrasound and imaging interpretation or the discriminatory capacity of current staging models. Accordingly, main research objectives include early detection of liver cancer in high-risk patients, accurate lesion classification to reduce diagnostic ambiguity, individualized prognosis to refine treatment allocation, and prediction of treatment response and recurrence risk to guide personalized follow-up. The expected outcomes of these approaches include improved diagnostic sensitivity and specificity compared with standard modalities, enhanced patient stratification beyond BCLC or TNM staging, and actionable insights for clinical decision-making. To ensure a proper evaluation and demonstrate the potential role of AI over standard-of-care approaches, studies use various experimental parameters such as performance metrics (AUC, sensitivity, specificity, concordace index) and validation strategies (internal cross-validation, external cohort testing, or prospective trials). These parameters are critical to ensure robust evaluation and to demonstrate whether AI tools truly add value over standard-of-care approaches.

### 3.1. Role of AI in Early Detection of Liver Cancer

Among the most pressing challenges in hepatology today remains the early detection of liver cancer—a shared goal that has long mobilized researchers and clinicians alike. Despite our collective efforts to improve diagnostic tools and enhance patient outcomes, several critical gaps remain. These range from limited awareness of screening protocols among physicians and patients, to the suboptimal use of ultrasound and other imaging techniques, and the lack of clarity around when and how to incorporate molecular biomarkers such as alpha-fetoprotein (AFP) or the GALAD (Gender, Age, AFP, AFP-L3, des-gamma-carboxyprothrombin (DCP)) score into clinical practice [20].

In this context, AI is emerging as a powerful ally—one that could transform how we manage patients with liver cirrhosis by supporting personalized screening strategies and more accurate risk assessment [21]. It is increasingly clear that not all individuals with advanced liver fibrosis (AF) share the same risk of developing liver cancer; conversely, liver cancer does not arise exclusively in those with AF. These nuances highlight the need for smarter, data-driven stratification tools [20]. Recent studies suggest that AI-based models can predict the likelihood of liver cancer development with impressive precision, offering new opportunities to move from generalized protocols to truly individualized surveillance. Liang et al. developed a CNN-based deep learning model that integrated clinical, imaging, histopathological, and molecular data to estimate the one-year risk of HCC, achieving an AUC of 0.94 in a cohort of 47,945 individuals (9553 with HCC) [22]. CNNs, which mimic human visual processing by steadily learning image features from simple to complex structures, were used to increase the accuracy of liver tumor identification on radiologic scans. Singal et al. applied both regression and machine learning models to predict HCC in 422 patients with Child A or B cirrhosis, using HALT-C data for validation. The ML approach, incorporating clinical and laboratory variables (e.g., AST, ALT, AFP, albumin), outperformed traditional methods [23]. Konerman et al. used longitudinal HALT-C data and applied ML algorithms (random forest, boosting), finding improved predictive performance for fibrosis progression and HCC, with a negative predictive value of 94% [24]. Ioannou et al. applied a deep learning RNN to 48,151 patients with HCV-related cirrhosis, showing superior performance to linear regression in identifying high-risk individuals [25]. Dawuti et al. combined urine-based SERS with an SVM classifier to distinguish cirrhosis and HCC based on metabolic profiles, achieving greater sensitivity than serum AFP [26].

The first step toward early diagnosis is early recognition. In patients with liver cirrhosis, ultrasound remains the cornerstone of initial imaging assessment. However, its diagnostic performance can vary significantly, as it is highly operator-dependent, with outcomes influenced by the experience and skill of the examiner [27]. To detect a nodule at an early stage, one must first be able to see it clearly—and this is not always straightforward. This is precisely where AI could play a transformative role. By enhancing image interpretation, AI may help clinicians better visualize hepatic nodules or suspicious lesions in patients with cirrhosis or advanced fibrosis, potentially identifying abnormalities that might otherwise go unnoticed [28]. In doing so, AI could support earlier detection and timely intervention—crucial steps in improving prognosis for liver cancer.

In cirrhotic surveillance, ultrasound alone detects early-stage HCC with a sensitivity of ~47% (95% CI 33–61%), improving to ~63% when AFP is added, but with reduced specificity; pooled sensitivity for detecting any-stage HCC is ~84% for both ultrasound and CT/MRI in surveillance contexts [27]. Against this benchmark, AI models have demonstrated significant promise. Yang et al. developed and externally validated a deep convolutional neural network (DCNN) using ultrasound images from 13 hospital systems, resulting in the “DCNN-US” model [29]. This AI tool outperformed experienced radiologists in both sensitivity and specificity, achieving an AUROC of 0.92 in distinguishing benign from malignant liver lesions—comparable to contrast-enhanced CT (84.7%) and slightly below MRI (87.9%).

In a more recent study from Thailand, the YOLOv5 deep learning model was trained on over 26,000 ultrasound images to detect and classify seven types of focal liver lesions (FLLs), including HCC and regenerative nodules [30]. The overall detection rate was 84.8%, with particularly high sensitivity for cholangiocarcinoma (92.2%) and HCC (82.3%). For regenerative nodules, specificity and negative predictive value reached nearly 100%. In practice, the YOLOv program passes the entire image through a deep neural network, which produces a high-level feature map. That map is then divided into a grid, like a checkerboard. Each square of the grid is responsible for spotting objects whose center lies inside it. How strict the model depends on two settings: the confidence threshold and the IoU threshold (how much overlap is allowed between boxes before they are considered duplicates). In this study, a confidence level of 0.25 and an IoU of 0.3 gave the best balance between diagnosing as many lesions as possible and avoiding too many false positive results.

Given the superior diagnostic accuracy of contrast-enhanced ultrasound (CEUS) compared to conventional B-mode ultrasound [31], combining AI with CEUS may further enhance lesion characterization. Guo et al. proposed a two-stage, multi-view learning framework based on selected CEUS images from the arterial, portal, and late phases [32]. This model effectively differentiated benign from malignant liver tumors and demonstrated the feasibility of CEUS-based computer-aided diagnosis. Together, these findings highlight the potential of AI to support clinicians in detecting and classifying liver lesions using both standard and contrast-enhanced ultrasound. Nevertheless, broader validation is essential before integration into routine practice.

### 3.2. Role of AI in Diagnosis of Liver Cancer

Once a nodule is identified in a patient with advanced cirrhosis, the key clinical challenge becomes determining its nature: is it benign, HCC, or intrahepatic cholangiocarcinoma? In many cases, a definitive diagnosis can be made non-invasively, particularly when imaging criteria on CT or MRI clearly indicate features typical of HCC [33]. However, a reliable diagnostic biomarker for HCC still eludes clinical practice. Perhaps it is not that such a marker does not exist—but rather that we have not yet learned how to see it. Here, artificial intelligence may offer a new perspective. By analyzing complex patterns that remain invisible to the human eye, AI could uncover signals we have been missing for decades. While current data are limited, the technology is already within reach—we now need time, validation, and vision to put it to work. The potential of neural networks in diagnosing HCC in cirrhotic patients was first explored by Poon et al. in 2001 [34]. By combining serum biomarkers (e.g., AFP, A1AT, A2MG, TBG, transferrin, albumin) with demographic data, their model improved sensitivity from 60% (AFP alone) to 73.8% (*p* < 0.05), with 88.2% specificity. Later, Sato et al. developed a machine learning tool with a user-friendly interface, integrating routinely available biomarkers to predict HCC, achieving an AUC of 0.94 [35].

Current SOC diagnosis relies on LI-RADS imaging, which yields pooled sensitivity and specificity of 0.86 and 0.85, respectively, for non-invasive HCC diagnosis [6]. By comparison, AI has delivered superior results in retrospective settings: Wang et al. trained an artificial neural network using serum protein fingerprints via mass spectrometry. Their model achieved 100% accuracy in distinguishing HCC from healthy individuals and 88.2% sensitivity when differentiating HCC from cirrhosis [28]. Similarly, Ksiazek et al. compared 10 machine learning algorithms using features like AFP, AST, GGT, and alkaline phosphatase [36]. The support vector machine (SVC) with feature selection performed best, achieving an accuracy of 88% [35]. These promising results support the integration of AI—especially neural networks—into early HCC detection workflows, emphasizing the value of combining multiple biomarkers with advanced modeling techniques.

Not all liver nodules can be confidently classified as HCC based solely on CT or MRI findings. Likewise, not every HCC displays the typical vascular enhancement pattern required for a non-invasive diagnosis. In such cases, liver biopsy remains the gold standard for establishing a definitive diagnosis in clinical practice [33]. However, AI may help reduce reliance on biopsy by improving the characterization of indeterminate nodules. By detecting subtle imaging features beyond human perception, AI could increase diagnostic confidence and expand the proportion of patients for whom a non-invasive diagnosis is sufficient—ultimately sparing them from the potential risks and complications of liver biopsy [37]. To improve clinical decision-making for cirrhotic patients with indeterminate liver nodules, Mokrane et al. developed a radiomics signature based on 13,920 CT-derived features from 178 patients. Using machine learning, the model achieved high accuracy in differentiating HCC from non-HCC lesions [38]. The study uses a radiomics-based machine learning model that relies on one specific feature—the change in image texture between the arterial and portal venous phases of a CT scan (referred to as “ΔV-A DWT1 LL Variance-2D”)—to diagnose liver tumors in patients with cirrhosis, reaching an AUC of 0.70 in the main group and 0.60 in the validation group.

More recently, Ho Yu et al. introduced four deep learning models for CT-based HCC diagnosis, selecting the Spatio-Temporal 3D Convolutional Network (ST3DCN) as the most effective. This model outperformed standard radiological assessments in terms of diagnostic AUC [39]. While AI applications in MRI remain limited due to technical complexity, preliminary results are promising. Hamm et al. developed a CNN-based system achieving 92% accuracy, 92% sensitivity, and 98% specificity for liver lesion classification [40]. Zhen et al. trained CNN models on MRI and clinical data from over 1200 patients. When unenhanced images were combined with clinical variables, the model reached an AUC of 0.985—matching the diagnostic performance of experienced radiologists [41].

#### AI Applications in Liver Cancer Pathology

The application of artificial intelligence in the pathological diagnosis of liver cancer has expanded significantly in recent years. AI tools, particularly those based on deep learning, have been applied to several diagnostic tasks including tumor detection, histological grading, subtyping, and biomarker quantification.

(1) Tumor detection and segmentation

One of the earliest and most well-established applications of AI in liver pathology is the detection of tumor regions within whole-slide images (WSIs). Deep learning models trained on annotated datasets can localize hepatocellular carcinoma foci, even in the presence of necrosis, fibrosis, or background steatosis. For example, Beaufrère and colleagues employed a weakly supervised deep learning model to automatically distinguish tumor from non-tumor regions in routine H&E-stained liver biopsies, demonstrating high agreement with expert pathologists [42]. These algorithms often use patch-level classification followed by aggregation to produce heatmaps that visually outline tumor areas, assisting in both diagnosis and tumor burden assessment.

Moreover, AI has proven effective in delineating tumor boundaries and detecting micro-invasive features. This is particularly relevant for surgical margin evaluation and in assessing intra-tumoral heterogeneity, which are critical for prognosis. In a comprehensive review, Calderaro et al. [43] highlighted that AI-based models can extract spatially resolved features that are invisible to the naked eye, enabling the mapping of subtle transitions between tumor and adjacent liver tissue.

(2) Histological grading and subtyping

Grading of HCC, based on the degree of cellular and architectural differentiation, plays a central role in predicting clinical outcome. However, grading remains prone to interobserver variability. Deep learning algorithms have been developed to classify HCC into well, moderately, or poorly differentiated subtypes with high reproducibility. A notable example is the study by Chen et al. [44], which used a Squeeze-and-Excitation Network (SENet) model to classify HCC grades from digitized pathology slides with an accuracy exceeding 95%, outperforming conventional CNN architectures.

Subtyping of liver tumors, particularly distinguishing HCC from intrahepatic CCA or cHCC-CCA, poses significant diagnostic challenges. Beaufrère et al. [42] demonstrated that weakly supervised clustering could identify histological features typical of HCC and intrahepatic CCA, supporting the differentiation of these entities and improving the recognition of cHCC-CCA. These results emphasize the potential of AI to assist in classifying ambiguous or poorly differentiated liver tumors, where morphological overlap confounds traditional diagnostic approaches.

In a prospective study evaluating the real-world impact of AI on liver cancer diagnosis, Kiani et al. developed a deep learning-based diagnostic assistant aimed at helping pathologists distinguish between HCC and CCA on H&E-stained WSIs [45]. The model, trained on image patches from The Cancer Genome Atlas (TCGA), achieved a diagnostic accuracy of 88.5% on an internal validation set and 84.2% on an external test set of 80 slides from Stanford University Medical Center. To assess the model’s clinical utility, the authors conducted a crossover study involving 11 pathologists of varying expertise levels who diagnosed the same cases with and without AI assistance. Although the AI assistant alone did not outperform the pathologists overall, its use significantly improved diagnostic accuracy for a subgroup of nine pathologists with well-defined experience levels (gastrointestinal subspecialists, non-GI specialists, and trainees; *p* = 0.045). Notably, when the AI model’s prediction was correct, pathologists’ accuracy improved markedly (*OR* = 4.28), but when the model was incorrect, diagnostic performance decreased significantly (*OR* = 0.25), highlighting the risks of automation bias. The study underscores that AI can serve as an effective augmentative tool in complex liver cancer diagnostics but also warns that incorrect AI outputs may mislead even experienced subspecialists. As such, the authors advocate for the thoughtful integration of AI, emphasizing transparency, user training, and real-time interpretability (e.g., heatmaps) in AI-assisted workflows. This study is among the first to demonstrate the nuanced interplay between AI assistance and human expertise in the pathological diagnosis of primary liver cancers.

Jang et al. developed a deep learning-based diagnostic pipeline for the histopathological classification of HCC, CCA, and metastatic colorectal cancer using hematoxylin and eosin-stained WSIs [46]. Utilizing a sequential two-step classification approach, their model achieved exceptional performance on internal datasets, with AUC values of 0.998 and 0.995 for distinguishing HCC from other liver cancers and CCA from metastatic colorectal cancer, respectively. However, testing on external datasets revealed limited generalizability (AUC = 0.745), which was subsequently resolved by incorporating external data into the training set, resulting in perfect classification (AUC = 1.000). These findings underscore both the promise and limitations of AI models in liver cancer diagnosis, highlighting the necessity for diverse training cohorts to ensure clinical robustness.

(3) Biomarker quantification and immunohistochemistry (IHC)

AI models have also been employed to quantify biomarker expression in liver tumor tissues, particularly in IHC. Traditional IHC evaluation is semi-quantitative and subject to interobserver variability. AI-driven image analysis can provide objective and reproducible quantification of markers such as Ki-67, glypican-3, PD-L1, and CD34. As highlighted by Mansur et al. [47], the use of AI in biomarker analysis facilitates not only diagnostic precision but also therapeutic decision-making, particularly in the context of immunotherapy eligibility.

Furthermore, multiplex IHC and spatial transcriptomics are being explored in conjunction with AI to assess the tumor immune microenvironment and molecular subtypes. Cifci et al. reviewed several approaches where AI-derived morphometric and molecular data from pathology slides were used to stratify tumors based on their immune infiltration profiles and predicted response to therapy [48].

### 3.3. AI for Staging and Prognosis of Liver Cancer

Several AI algorithms have been developed lately to facilitate diagnosis of liver malignancy on radiological images; however, research is much scarcer when it comes to tumor staging. Z. Zhao et al. developed a model based on the records of 998 patients with HCC, which incorporates 112 patient variables, including demographics, histology, laboratory test results, and past medical history, and classifies patients into four staging groups with significant survival differences between each stage, far more distinct than the BCLC system [49]. However, the model lacks external validation and lacks any genetic data, which authors themselves acknowledge could prove significant in relation to HCC staging and prognosis. Another AI tool using deep learning radiomics with data extracted from computer tomography images was developed by Y. Wang et al. with the purpose to evaluate regional lymph node involvement in preoperative assessment of patients with hilar cholangiocarcinoma, with 14.6% higher accuracy in diagnosis when compared to radiologists [50]. The study uses a deep learning radiomics model that extracted image features from CT scans which were fed in an SVM plus three key clinical factors to assess lymph node involvement before surgery in patients with hilar cholangiocarcinoma. This hybrid approach achieved strong accuracy, with an AUC of about 0.87 for detecting any lymph node metastasis and up to 0.95 for distinguishing early-stage (N1) from more advanced metastasis (N2).

Application of AI algorithms for the development of prognostic systems which help guide treatment strategies has been the focus of many studies in the past few years, due to the fact that traditional frameworks such as BCLC and TNM staging for HCC remain widely used but are limited in granularity. D. Guo et al. [51] developed a radiomics-based model using data from preoperative CT scans to accurately predict HCC recurrence after liver transplantation. Microvascular invasion (MVI) at the time of liver resection for hepatocellular carcinoma is a major determinant of early HCC recurrence and survival and has been the focus of many studies with the purpose of developing models to predict preoperative MVI, which are summarized in Table 1. Among the articles underlined, Wei et al. [52] performed prospective external validation to assess the models developed, while Wang et al. [53], Xia et al. [54], and Zang et al. [55] had external validation. For MVI—a key prognostic factor—deep learning models demonstrate pooled AUCs of 0.90, exceeding radiologists’ visual assessment and informing surgical strategy. Numerous AI algorithms have also been developed with the purpose of predicting response to locoregional treatments such as TACE, transarterial radioembolization (TARE), or microwave/radiofrequency ablation (MWA/RFA) in patients with hepatocellular carcinoma, summarized in Table 2. These studies underline the importance of AI models in predicting prognosis after liver transplantation, after liver tumor resection, or predicting response to locoregional therapies in patients with HCC. More studies are needed to improve the performance of such models and to eventually integrate them into clinical practice.

### 3.4. AI for Treatment Decision

Treatment decision-making in liver malignancies remains complex, due to heterogeneous tumor biology and competing therapeutic options. Traditional staging systems (such as Milan criteria) offer general guidance but often fail to account for individual variability [7,77,78]. AI has emerged as a powerful tool to support personalized treatment strategies by integrating data from imaging, histopathology, molecular profiles, and clinical features. In HCC, AI assists in selecting candidates for TACE, resection, or transplantation; in iCCA, it refines surgical planning and recurrence prediction; and in hepatic metastases, it helps predict response to systemic or local therapies. These applications reflect a growing shift toward AI-enabled precision in liver cancer management.

Treatment decision-making for HCC lies at the crossroads of oncology, hepatology, and liver surgery. It often requires a delicate balance between tumor burden, underlying liver function, and patient comorbidities—factors that may vary significantly between individuals. Although the BCLC staging system is widely adopted for therapeutic stratification [7], it has well-recognized limitations in capturing the biological heterogeneity and real-world variability of HCC patients [79,80]. AI, particularly through ML and DL approaches, has shown promise in complementing traditional frameworks by integrating complex clinical, imaging, and molecular data. These models offer the potential to refine prognostic stratification, improve treatment selection, and support more individualized care pathways—especially in cases where guidelines leave room for clinical judgment [81,82,83]. As such, AI is not replacing clinician expertise but augmenting it with tools capable of uncovering patterns beyond human perception.

One of the earliest and most relevant applications of AI in treatment guidance is risk stratification and survival prediction, which can indirectly influence therapeutic choices. In a large-scale study, Liu et al. developed and validated an artificial neural network (ANN) model to predict 1-year progression-free survival (PFS) in patients with hepatitis B-related HCC, using data from nearly 2900 individuals. Their ANN integrated a combination of clinical and biological features—such as tumor size, AFP level, and immune parameters like CD4 T-cell count—and outperformed conventional systems (e.g., BCLC, TNM, CLIP), achieving an AUROC of 0.866 for PFS prediction. This stratification can meaningfully inform clinicians whether curative intent treatment (resection, ablation) is justifiable or whether a more conservative or systemic approach is warranted [81].

Beyond prognostication, AI has been increasingly applied to guide therapeutic decisions directly, particularly in early-stage HCC. Li et al. utilized three distinct ML models—DeepSurv, random survival forests (RSF), and neural multi-task logistic regression (N-MLTR)—trained on SEER data to generate personalized treatment recommendations regarding liver resection or transplantation. Not only did the ML models outperform traditional Cox models in survival prediction (e.g., C-index for DeepSurv: 0.7028), but patients who followed the ML-suggested treatment path had significantly improved survival. These findings highlight the capacity of AI to tailor treatment to individual prognostic profiles, potentially serving as adjuncts in multidisciplinary decision-making [82].

Imaging-based AI models have also demonstrated value in refining therapeutic strategies. Schön et al. conducted a comparative analysis between radiomics and deep CNN to predict overall survival from pre-treatment CT scans across different HCC stages and treatment modalities. CNN models achieved superior robustness (C-index up to 0.71) compared to radiomics alone (C-index as low as 0.51), underscoring the potential of DL to extract latent features from imaging that may relate to tumor biology, vascularity, or treatment responsiveness. Such models are especially valuable in intermediate-stage disease, where TACE candidacy must be balanced against early initiation of systemic therapy [83]. Another critical application is the non-invasive prediction of MVI, a key prognostic factor influencing surgical strategy. A meta-analysis by Zhang et al. pooled data from 16 studies (*n* = 4759 patients) and demonstrated that deep learning models achieved an AUC of 0.90 for preoperative MVI prediction—superior to traditional radiologic assessment and even to other non-deep learning models. This predictive power allows for early identification of high-risk patients, informing decisions such as whether to favor anatomical resection over ablation, or to intensify surveillance postoperatively [84]. Finally, radiomics using high-resolution MRI has shown promise in guiding treatment by predicting response, recurrence risk, and tumor aggressiveness. Xie and Chen highlighted the growing role of MRI-based radiomics in preoperative planning, particularly through the prediction of features such as microvascular invasion, tumor grade, and treatment response. The ability to extract these features from imaging prior to intervention can support personalized therapeutic pathways [85].

Emerging approaches extend beyond imaging and genomics; for example, machine learning has been applied to optimize traditional medicine-based interventions in liver disease, such as the precision strike strategy using Xiao-Chai-Hu Decoction, suggesting opportunities for integrative AI frameworks in HCC [86].

AI has also begun to reshape the transplant landscape. Pinto-Marques et al. introduced HepatoPredict, a gene expression-based ML algorithm combining four tumor biopsy biomarkers with clinical variables to assess post-transplant recurrence risk. The model successfully identified 99% of disease-free patients at 5 years in retrospective cohorts, with external validation even capturing 16–24% of patients outside Milan criteria who would have been excluded under current guidelines. This suggests that AI models could refine transplant eligibility beyond morphological criteria, improving allocation in settings of organ scarcity [87].

Taking together, current evidence indicates that AI-based models can augment clinical decision-making in HCC by predicting outcomes, guiding treatment selection, and identifying patients most likely to benefit from specific modalities. However, real-world implementation remains limited by data heterogeneity, lack of prospective validation, and concerns about explainability.

### 3.5. AI in Monitoring, Recurrence Detection of Liver Cancer

Accurate detection of liver cancer recurrence is mandatory, and AI holds promise in the reappraisal of the patients and the disease status. The follow-up of a liver cancer patient entails various management strategies, contingent upon the type of treatment received. Tumor recurrence prediction has been a thoroughly studied subject, with many advancements in this field aiding clinicians in monitoring patients. Regardless, a daunting challenge to prevent liver cancer recurrence remains, as patients, even the ones who undergo curative intent treatments, still present with a recurrence rate that exceeds 50% [88]; this high rate is a major contributor for the poor prognosis of patients, and therefore improved identification of reliable recurrence predictors could significantly optimize surveillance strategies for the liver cancer patient.

For instance, a machine learning model was trained to predict early recurrence (ER) (under 12 months after treatment) in 536 ICC patients who underwent surgical resection; the model used 14 clinical and pathological characteristics and led to the development of an easy to use calculator that could be implemented in clinical practice in order to personalize management strategies for patient follow-up [89]. In their work, Alaimo et al. developed and validated machine learning models to predict early recurrence of intrahepatic cholangiocarcinoma after curative hepatectomy. They utilized a dataset of 536 patients drawn from an international database, randomly splitting the sample into a training cohort (70%, *n* = 376) and a testing cohort (30%, *n* = 160) for rigorous validation. The random forest model achieved the best predictive performance, with an AUC of 0.904 in the training set and 0.779 in the independent testing set, demonstrating solid generalization performance.

Several recent studies have revealed the significance of machine learning models for HCC recurrence prediction [69,90,91,92]; the only drawback is that most studies rely on a predefined set of manually engineered features that help predict liver cancer recurrence. In contrast, deep learning algorithms can automatically identify specific imaging features that are indicative of HCC recurrence, without the need for human interference. Such is the case with a study conducted by Kucukkaya et al. [93] who used a deep learning tool that inferred post-treatment early-stage HCC recurrence from pre-treatment MRI imaging. The study demonstrated superior performance for predicting early liver cancer recurrence compared to conventional models.

Radiomics, which involves the extraction of quantitative features from medical images, combined with AI techniques, has also shown potential in predicting HCC recurrence. A deep learning-based radiomics approach using CEUS was developed to predict PFS and guide treatment selection between RFA and surgical resection in very-early and early-stage hepatocellular carcinoma [94]. In a retrospective cohort of 419 patients, radiomics models and nomograms accurately stratified 2-year PFS and identified patients who would benefit from switching treatments. This strategy improved predicted outcomes by 12–15% in selected cases. The results suggest that radiomics-guided decision-making could enhance personalized treatment planning in early-stage HCC. For HCC, standard-of-care follow-up relies on fixed-interval CT/MRI and AFP measurement. AI enables risk-adaptive monitoring; a CT-based radiomics signature was developed to predict early recurrence (≤1 year) of HCC after surgery—Zhou et al. [94] revealed that radiomics and deep learning models can predict early recurrence with AUCs between 0.82 and 0.98 after surgery, ablation, or TACE. In a cohort of 215 patients, the radiomics model outperformed clinical models alone, achieving an AUC of 0.817. Combining radiomics features with clinical variables significantly improved the accuracy of preoperative risk assessment for early recurrence. By incorporating variables such as tumor size, vascular invasion, and serum alpha-fetoprotein levels, these models provide a comprehensive assessment of recurrence risk, with an independent external cohort being used to validate the results [95]. Other studies have also utilized machine learning models to analyze radiomic features from preoperative imaging, enabling risk stratification of patients based on their likelihood of recurrence. Such predictive modeling can inform personalized surveillance schedules and guide clinical decision-making [96,97].

AI has also shown promise in identifying biomarkers associated with HCC recurrence. Deep learning models trained on multi-omic datasets—including RNA sequencing, miRNA profiles, and DNA methylation—have been used to classify aggressive HCC subtypes linked to poorer outcomes and higher recurrence risk. Notably, one such model identified subtypes characterized by frequent TP53 mutations and upregulated markers like KRT19 and EPCAM [98].

Through an extensive review of global databases [99], over 300 studies were found applying AI to predict recurrence after liver cancer treatment, primarily in HCC cases treated with PA, SR, or TACE. A summary of the most important studies is depicted in Table 3. Among them, hybrid radiomics–deep learning approaches yield superior performance by capturing complex imaging features linked to recurrence risk. Despite the fact that machine learning models prove to be valuable, predominantly when combining them with clinical data, the most promising track appears to be multimodal frameworks that integrate clinical, imaging, and molecular data to deliver precise and clinically relevant risk stratification. Additionally, from all the articles mentioned, Yamashita et al. [100], Lai et al. [101], Zhang et al. [102], Zeng et al. [103] had external validation once again contouring the early stages of AI research. Regarding ICC research, there is a scarcity of data for PA or TACE treatment, with only limited work after SR. Radiomics and deep learning dominate, with pathomics emerging but still underdeveloped. These findings highlight both the progress and gaps in AI-driven recurrence prediction across liver cancer subtypes.

An ideal staging approach for recurrent liver cancer should integrate comprehensive, disease-specific data, including tumor burden, recurrence patterns, liver function, viral activity, and prior treatments. AI can streamline this process by analyzing complex multimodal inputs to support accurate prognostic stratification and personalized treatment allocation. By uncovering subtle patterns in clinical, imaging, and pathological data, AI may enhance our understanding of recurrent HCC and inform the development of more precise staging systems.

Recent studies show that no single AI method dominates across all applications in liver cancer care; instead, performance depends on the specific clinical task. In regard to liver cancer diagnosis, CNNs continuously perform best, being often superior to radiologists in sensitivity and specificity on MRI and B-mode ultrasound [29,40]. In treatment planning, ML and radiomics are prevalent [81,82,83]; the HepatoPredict biopsy gene signature is outstanding as an externally validated model for transplant selection [87]. For recurrence detection, the best performance metrics are reported as an AUC of 0.84 for MRI radiomics nomograms [89], and a concordance index of 0.77 for CT radiomics [95]. Looking ahead, future research should focus on multicenter studies to validate AI models across diverse populations and clinical settings. The development of standardized protocols for data collection and analysis will be crucial in facilitating the adoption of AI in routine clinical practice.

## 4. Barriers to Implementation—Challenges and Future Directions

Despite the rapid development of AI and the hype surrounding its potential use for augmenting the clinical process of a liver cancer patient, from diagnosis to treatment and follow-up, its implementation has been hindered by several barriers. Most research so far focuses on the development and evaluation of advanced technical AI-based processes, but the gap between their potential and their clinical deployment persists [110]. A framework that ensures AI is effectively applied in everyday clinical workflows—both to leverage its benefits and to recognize its limitations, is still lacking.

### 4.1. Liability

One of the primary and critical problems that need to be addressed concerns liability: AI implementation of any kind requires government and other regulatory bodies to create protocols; additionally, the healthcare provider needs to address the obvious complication of over reliance on AI results while the developer needs liability to cover himself for inaccurate predictions [111]. Due to the rapid development of AI, policies have not kept the pace, thus leaving healthcare providers without specific regulations regarding accountability when using AI: who takes the blame if an AI-specific code makes a medical decision that was the wrong one for the patient? Furthermore, their exact role in the patients’ medical journey needs to be established: is AI a helping tool for the physician or can it make independent decisions? In order to answer these questions, AI systems need to have transparent algorithms that help physicians clearly understand how a decision was made. The “black box” nature of AI is a barrier for clinicians due to the lack of understanding the decision-making model; trust can be fostered through implementation of explainable AI techniques. Leaders in decisional forums can also impede or contribute to overcoming these challenges and change or implement certain laws to develop a liability insurance that covers both the software developer and the healthcare provider and provides a comprehensive legal–ethical framework [112].

### 4.2. Education

The digitalization of the healthcare system is rapidly expanding while the education process of healthcare professionals is still lacking. Educational programs need to be implemented from a student level to have trained staff that understand and can maximize the potential of an AI program. The lack of randomized controlled trials prior to real-world deployment is also a risk for causing patient harm, thus undermining trust in the medical profession and concluding in a flawed perception of the public regarding AI benefits [113]. The implementation of AI is contingent upon engaging the community (from scientist to clinician to the patient involved), therefore accessibility to information regarding the potential benefits and limitations needs to be encouraged through various platforms, like research papers, talks at conferences, or at patient’s associations [114].

### 4.3. Ethical Barriers

Cybersecurity measures need to address the concepts of data sharing and patient confidentiality. The European Union clearly states thorough the General Data Protection Regulation (GDPR), that the patients “own and control their own data and must give explicit consent for its use or when it is shared” [115]. A recent review pointed out that there are a significant number of articles that address concerns about the improper use of clinical data; patients need to be properly informed about the use, storage, and potential access to their personal information in AI programs [116,117]. Clear governance frameworks are essential for the implementation of AI in healthcare, particularly to define the roles and responsibilities of all stakeholders regarding data ownership and usage. A study by Marcu et al. emphasizes the critical need for robust data privacy policies to ensure the secure storage, use, and sharing of sensitive information, thereby minimizing the risk of confidentiality breaches [118].

### 4.4. Integration into Clinical Workflow

The variability of clinical data are another challenge for AI implementation. Data size and quality are crucial when it comes to AI system training: the challenge is to amalgamate high quality data that ensures proper training for machine learning algorithms. Many AI models are trained on small, biased, or retrospective datasets, limiting their generalizability and risking patient harm. For example, models trained primarily in Asian HCC cohorts often underperform in Western populations, where alcohol-related or NASH-related etiologies predominate [21]. The algorithmic bias stems from AI systems being trained on imbalanced or demographically narrow datasets which may underperform in underrepresented populations, thereby exacerbating health disparities. This is particularly concerning in hepatocellular carcinoma, where etiologic and demographic risk factors vary widely across geographic regions. Without standardized data, robust validation, and sufficient infrastructure, AI systems remain frail, underutilized, and difficult to scale safely [118]. Additionally, AI programs need to receive feedback to learn from their mistakes and improve their diagnosis performance; there is currently no program in place that ensures this process, therefore AI systems cannot accurately adapt and change to ensure high quality performance [119].

Second, there is a lack of external and prospective validation. A recent review called attention to the “generalizability” issue, the lack of one to be specific. The study revealed that there are only a few AI studies in hepatology that included external testing, and only a minority were prospective, stressing the need for data adjusting and the risk of overfitting and optimistic performance estimates, or underfitting [120].

Third, the “black box” nature of AI poses significant challenges for the clinician. Tools such as gradient-based methods, which highlight the exact part of a scan that influenced the AI’s decision, are being developed to improve explainability, but their integration into hepatology studies remains scarce [18].

Finally, an essential step toward real-world adoption is embedding AI tools into the multidisciplinary tumor board (MDT) setting, where treatment decisions for liver cancer are routinely made by a team comprising hepatologists, surgeons, radiologists, oncologists, and pathologists. Several studies on large language models have reported encouraging results regarding the use of AI as a support tool in the MDT setting [121,122,123,124]. In this framework, AI could assist as decision-support platforms by automatically combining clinical data with imaging, pathology, or molecular data, and presenting risk scores or treatment response predictions in a format that is transparent for all team members.

### 4.5. Social Barriers

Bias in training datasets can lead to biased performance across patient groups, emphasizing existing health disparities. Underrepresented populations may receive less accurate predictions; there is a fine line between recognizing differences and generating discrimination and AI systems do not possess the subtlety needed to ascertain if these differences have a causative effect on the outcome [125]. Furthermore, digital exclusion and variable access to AI tools can create a two-tiered healthcare system, while public skepticism and preference for human interaction hinder acceptance [126].

## 5. Conclusions

Looking ahead, a strategic roadmap that bridges the gap between recent advancements of AI and clinical adoption is required to advance AI in clinical care. To establish generalizability, studies with prospective validation are needed across diverse healthcare systems. Additionally, to develop personalized prognostic and therapeutic models, studies should focus on integration of clinical, imaging parameters with multi-omics data. In parallel, regulatory science and international collaboration are essential to guide model standardization, maintain continuous performance monitoring, and promote equity across healthcare settings. The outcome of this study is therefore twofold: first, to provide a comprehensive synthesis of the state-of-the-art AI applications for liver cancer, highlighting tangible clinical advantages and persisting shortcomings; and second, to propose a forward-looking roadmap that emphasizes prospective multicenter validation, multi-omics integration, explainable AI, and ethical/regulatory alignment as essential steps for clinical translation. By addressing these priorities, future research can move AI tools beyond proof-of-concept toward safe, effective, and equitable clinical adoption in liver cancer care. The time has come to move beyond promise, confront existing challenges, and help herald a new era of precision medicine for patients with liver cancer.

## Figures and Tables

**Figure 1 cancers-17-03003-f001:**
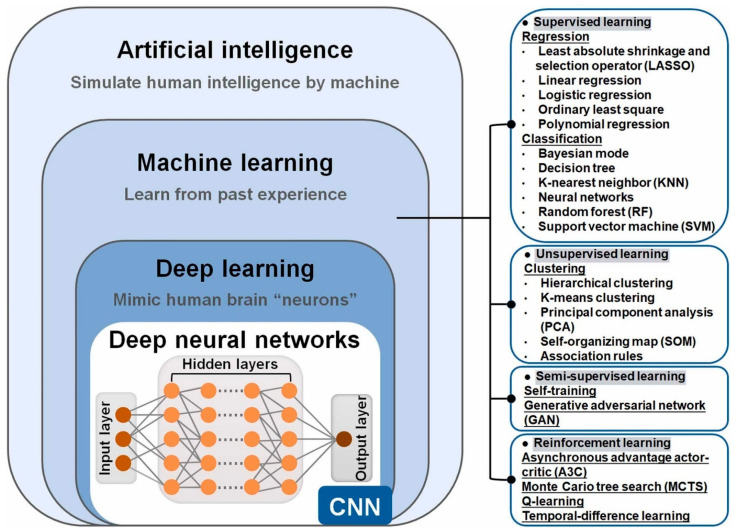
AI classification. Abbreviations: 1. CNN: Convolutional neural network. With permission from [12].

**Table 1 cancers-17-03003-t001:** Summary of studies evaluating artificial intelligence models based on imaging for the prediction of microvascular invasion in patients with HCC awaiting resection.

Author, Year (Ref)	AI Method	Number of Patients	Imaging Technique	Performance Metrics
Dong Y, 2020 [56]	radiomics	322	ultrasound (grayscale)	AUC 0.74 SEN 0.89 SPE 0.48
Xu X, 2019 [57]	radiomics	495	CT	AUC 0.88 SEN 0.88 SPE 0.76
Ma X, 2019 [58]	radiomics	157	CT	AUC 0.80 SEN 0.89 SPE 0.76
Liu Q-P, 2020 [59]	radiomics	494	CT	AUC 0.79 SEN N/A SPE N/A
Jiang Y-Q, 2020 [60]	radiomics CNN	405	CT	AUC 0.85 SEN 0.93 SPE 0.75
Song D, 2021 [61]	CNN, radiomics	601	MRI	AUC 0.93 SEN 0.88 SPE 0.88
Wang G, 2021 [62]	CNN	114	MRI	AUC 0.92 SEN 0.86 SPE 0.88
Y Zhang, 2021 [63]	CNN	237	MRI	AUC 0.72 SEN 0.55 SPE 0.81
Fu S, 2021 [64]	multi-task deep learning	366	CT	AUC 0.83 SEN N/A SPE N/A
Wei J, 2021 [52]	CNN	750	CT, MRI	MRI vs. CT AUC: 0.812 vs. 0.736 SEN 0.70 vs. 0.57 SPE 0.80 vs. 0.86
Yang Y, 2022 [65]	radiomics	283	CT	AUC 0.90 SEN 0.91 SPE 0.97
Wang F, 2023 [53]	CNN	397	CT + MRI	AUC 0.84 SEN 0.77 SPE 0.84
TY Xia, 2024 [54]	radiomics	773	CT	AUC 0.84 SEN 0.74 SPE 0.81
Z Zhou, 2023 [66]	DL, radiomics	140	CT	AUC 0.85 SEN 0.87
T Wang, 2023 [67]	DL	233	MRI	AUC 0.81 SEN 0.78 SPE 0.67
W Zhang, 2024 [55]	DL, radiomics	576	CEUS	AUC 0.73 SEN 0.60 SPE 0.76

Abbreviations: AUC—area under the curve, SEN—sensitivity, SPE—specificity, CNN—convolutional neural network, DL—deep learning, ML—machine learning, CEUS—contrast-enhanced ultrasound, CT—computed tomography, MRI—magnetic resonance imaging.

**Table 2 cancers-17-03003-t002:** Studies evaluating AI models for the prediction of response to locoregional therapies in patients with HCC.

Author, Year (Ref)	Aim	AI Method	Number of Patients	Imaging Technique	Performance Metrics
Morshid A, 2019 [68]	predict response to TACE	CNN	105	CT	Accuracy 0.74 SEN N/A SPE N/A
Shan Q, 2019 [69]	predict recurrence after curative resection or ablation	radiomics	156	CT	AUC 0.79 SEN N/A SPE N/A
Peng J, 2021 [70]	predict response to TACE	DL, radiomics	310	CT	AUC 0.99 SEN 0.93 SPE 0.94
Abajian A, 2018 [71]	predict response to TACE	machine learning	36	MRI	AUC 0.78 SEN 0.62 SPE 0.82
Oezdemir I, 2020 [72]	predict response to TACE	machine learning	36	CEUS	AUC 0.86 SEN 0.89 SPE 0.82
Liu D, 2020 [73]	predict response to TACE	DL, ML, radiomics	130	CEUS	AUC 0.93 SEN 0.89 SPE 0.92
Aujay G, 2022 [74]	predict response to TARE	radiomics	22	MRI	AUC 1 SEN 1 SPE 1
Pino C, 2021 [75]	predict response to TACE	DL	126	CT	AUC 0.81 SEN 0.83 SPE 0.82
Chen M, 2023 [76]	predict response to TACE	DL	144	MRI	AUC 0.79 SEN 0.71 SPE 0.85

Abbreviations: TACE—transarterial chemoembolization, CNN—convolutional neural network, CT—computed tomography, SEN—sensitivity, SPE—specificity, AUC—area under the curve, DL—deep learning, MRI—magnetic resonance imaging, ML—machine learning, CEUS—contrast-enhanced ultrasound, TARE—transarterial radioembolization.

**Table 3 cancers-17-03003-t003:** Summary of studies exploring the use of AI for liver cancer recurrence prediction.

Study	AI Method	Cancer Type	Perfomance Metrics
Huang et al. [104]	CEUS ultrasomics	HCC	AUC of 0.845
Lv et al. [105]	deep learning based radiomics	HCC	AUC of 0.98 in the training cohorts and 0.83 for the testing cohorts
Yamashita et al. [100]	CNN	HCC	concordance indices of 0.724 on internal cohorts and 0.683 on external cohorts
Peng et al. [70]	DL and random forest algorithm	HCC	AUC of 0.995 in the training cohort and of 0.994 in the validation cohort
Chicco et al. [106]	Machine learning	HCC	predict recurrence using prognostic indicators (ALP; AFP; hemoglobin)
Zhang L et al. [102]	Deep learning	HCC	predict recurrence after TACE with a 78.2 % accuracy for the internal cohort and 75.1% accuracy for the external cohort
Wang et al. [107]	Deep learning	HCC	predict overall survival after hepatectomy with AUCs of 0.8065, 0.7404, and 0.7944
Zhou et al. [108]	Deep learning	HCC	predict early recurrence with AUCs of the clinical and combined models of 0.781 and 0.836
Zeng et al. [103]	Random survival forest	HCC	predict early recurrence with a concordance index value of 0.725, 0.762 and 0.747 for the training, internal and external cohorts
Lai et al. [101]	Deep learning	HCC	predict post-transplant recurrence at 5 years with a concordance index of 0.78
He et al. [109]	Deep learning	HCC	predict post-transplant recurrence with an accuracy of 0.82%

Abbreviations: AI—artificial intelligence, HCC—hepatocellular carcinoma, CNN—convolutional neural network, TACE—transarterial chemoembolization, CEUS—contrast enhanced ultrasound, ALP—alkaline phosphatase, AFP—alpha-fetoprotein, AUC—area under the curve.

## Data Availability

No new data were created or analyzed in this study. Data sharing is not applicable to this article.

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
