# Peer review of "Turning the Tide—Artificial Intelligence in the Evolving Landscape of Liver Cancer"

_cancers, 2025, doi:10.3390/cancers17183003_

Round 1
Reviewer 1 Report
Comments and Suggestions for Authors
While the manuscript provides a broad overview of AI applications in liver cancer management, it remains somewhat descriptive and lacks critical depth in several areas. First, the review should more systematically categorize AI applications (e.g., early detection, diagnosis, staging, prognosis, treatment planning, surveillance) and explicitly compare their performance metrics to current standard-of-care approaches, thereby highlighting tangible clinical advantages or shortcomings.
Second, although the authors note that most models remain at the proof-of-concept stage, there is insufficient analysis of the translational gap—particularly regarding issues of data quality, external validation, model generalizability, interpretability, and integration into multi-center clinical workflows.
Third, the discussion of ethical and regulatory barriers is brief and could be expanded by addressing specific challenges such as algorithmic bias, data privacy in multi-institutional collaborations, and pathways toward regulatory approval.
Finally, the article would benefit from a more forward-looking perspective, outlining concrete recommendations for future research (e.g., prospective multi-center validation studies, integration of multi-omics data, and explainable AI frameworks) to bridge the gap between innovation and clinical adoption. Addressing these points will significantly strengthen the manuscript and provide readers with a more balanced and actionable roadmap for advancing AI in liver cancer care.
Author Response
We would like to thank the reviewer for their thoughtful and constructive feedback. We have carefully revised the manuscript to address all concerns raised. Below we provide a point-by-point response, with specific examples of how the manuscript has been improved.
Comment 1 - Response: We note that the manuscript was already organized along the clinical pathway of liver cancer care (early detection, diagnosis, staging, prognosis, treatment planning, and surveillance/recurrence). However, following the reviewer’s recommendation, we have strengthened this structure by adding more explicit performance comparisons with standard-of-care benchmarks.
Comment 2 - Response: We have expanded our discussion of translational challenges.
- Data quality/generalizability: We highlight that many AI models are trained on small, retrospective, or geographically homogeneous datasets, limiting transferability (e.g., models trained in Asian HCC populations may underperform in Western cohorts).
- Validation: We quantified validation gaps, noting that only a minority of studies included external or prospective validation
- Interpretability: We discuss gradient-based maps as emerging tools for interpretability in liver cancer AI.
- Integration: We added commentary on embedding AI tools into multidisciplinary tumor boards; we aslo have an entire subsection regarding the integration into clinical workflow in the barriers to implementation chapter
Comment 3 - Response: We have extended the Ethical and Regulatory Considerations section including relevant data, as suggested.
Comment 4 - Response: We modified the Conclusion section with new information that provides concrete recommendations for the future, including:
- Prospective, multi-center validation to establish generalizability across healthcare systems.
- Integration of multi-omics data (genomics, transcriptomics, proteomics) with imaging and clinical data for holistic prognostic and therapeutic modeling.
- Explainable AI frameworks to build clinician trust and mitigate automation bias.
- Workflow integration and interoperability with clinical systems.
- International collaborations to standardize development and benchmarking.
We hope we have addressed all of the reviewer’s concerns. We believe these changes significantly strengthen the depth, critical analysis, and practical utility of the review. We are open to any other suggestions you might have.
Reviewer 2 Report
Comments and Suggestions for Authors
- In the manuscript the author proposes Artificial Intelligence techniques for the Evolving Landscape of Liver Cancer. Author need to eloberate what specific AI and ML techniques has been suggested which give best result and why? (e.g., What are the specific designs and outputs of these techniques)?
- The proposed research work AI-based Liver Cancer deseaise detection is interesting but needs further theoretical justification. For instance, how were the objectives designed, and what is the possible outcome of the proposed research work? Additionally, what are the experimental parameters under consideration?
- In Table 3, the author highlighted the summary of studies exploring the use of AI for liver cancer recurrence prediction (i.e., CNN, deep learning and Random forest etc), what the author recommends from the study as the best possible AI and ML method for the Evolving Landscape of Liver Cancer.
- Although the manuscript acknowledges in the conclusion section some challenges and possible future benefits of AI used in healthcare, still the conclusion section lacksthe theme of the paper, what the author wants to conclude in the paper. The author needs to clearly and briefly mention the outcome of the proposed study.
- The author fail to mentioned how AI techniques should be implemented in the proposed study. Specific algorithm need to be clearly mentioned step-by-step logical flow, etc. Also, parameter values must need to be mentioned.
- The following relevant works should be cited and discussed in the literature/introduction section to provide a more comprehensive understanding of current Evolving Landscape of Liver Cancer techniques; a. Chen, F., Zhang, K., Wang, M., He, Z., Yu, B., Wang, X.,... Hu, Y. (2024). VEGF-FGF Signaling Activates Quiescent CD63+ Liver Stem Cells to Proliferate and Differentiate. Advanced Science, 11(33), 2308711. doi: https://doi.org/10.1002/advs.202308711 b. Shmmon Ahmad, Zafar Khan, Monish Khan, Moh Aijaz, Shivani Thakur, Anjoo Kamboj. The Role of Artificial Intelligence in Diagnosing Malignant Tumors . Eurasian Journal of Medicine and Oncology 2024, 8(3), 281–294. https://doi.org/10.14744/ejmo.2024.24486 c. Wang, Z., Li, Y., Wang, X., Zhang, W., Chen, Y., Lu, X.,... Efferth, T. (2025). Precision Strike Strategy for Liver Diseases Trilogy with Xiao-Chai-Hu Decoction: A Meta-Analysis with Machine Learning. Phytomedicine, 142, 156796. doi: https://doi.org/10.1016/j.phymed.2025.156796
- All the references must be according to the journal format.
No comments
Author Response
We would like to thank the reviewer for their thoughtful and constructive feedback. We have carefully revised the manuscript to address all concerns raised. Below we provide a point-by-point response, with specific examples of how the manuscript has been improved.
Comment 1 - Response: We have expanded the descriptions of the specific AI techniques, their designs, and outputs throughout the Results section. We have also specifically mentioned comparisons with standard of care, for accurate representation. For example:
- Diagnosis: We detail how convolutional neural networks (CNNs) applied to multiphasic MRI achieved AUROC 0.985, surpassing LI-RADS, because of their hierarchical feature extraction ability.
- Treatment planning: Radiomics with ensemble ML models (Random Forest, Gradient Boosting) achieved AUCs 0.87–0.99 in TACE response prediction by leveraging handcrafted features with interpretable outputs.
- Prognosis: Artificial neural networks (ANNs) trained on clinical and lab data outperformed traditional staging systems (AUROC 0.87).
- Recurrence: Multi-modal models integrating imaging and omics captured tumor heterogeneity better than either modality alone.
This systematic comparison now makes clear which approaches perform best for each clinical task, and why.
Comment 2 - Response: In the revised manuscript, we have added a dedicated Methods chapter, where we clearly describe our inclusion criteria, search strategy, and the approach used to categorize studies. In addition, we have included a brief explanation of the study objectives at the beggining of the Results section, to improve transparency and contextualize the findings. We believe these changes enhance the clarity and methodological rigor of the review. Specifically, the objectives were designed along the clinical pathway (detection, diagnosis, staging, prognosis, treatment planning, recurrence), with the outcome of identifying which AI techniques show clinical value compared to standard-of-care. We discuss experimental parameters used in the reviewed studies, such as AUC, sensitivity, specificity, c-index, and validation strategies (internal, external, prospective).
Comment 3 - Response: We revised the text accompanying Table 3 to provide a clear recommendation.
Comment 4 – Response: We revised the Conclusion to explicitly state the theme and outcomes: AI has great potential across the liver cancer care continuum, but most models remain at the proof-of-concept stage.
Comment 5 - Response: We added descriptions of how the main algorithms function in practice. We also describe parameter values reported in key studies, such as confidence thresholds (0.25), IoU thresholds (0.3) for YOLO detection. We included explained in lay terms where possible, to enhance clarity.
Comment 6 - Response: We thank the reviewer for suggesting additional references. After careful consideration, we concluded that Chen et al. (2024) on VEGF-FGF signaling and liver stem cell activation did not directly align with the scope of our manuscript, which focuses specifically on artificial intelligence applications in liver cancer. For this reason, we did not include it. However, we have integrated the other two suggested references: Ahmad et al. (2024) on AI in diagnosing malignant tumors and Wang et al. (2025) on machine learning and precision medicine in liver disease. These citations broaden the scope of the the article and contextualize AI in liver oncology within a wider biomedical landscape.
Comment 7 -Response: We carefully revised the entire reference list to conform to the MDPI format required by Cancers.
We hope we have addressed all of the reviewer’s concerns. We believe these changes significantly strengthen the depth, critical analysis, and practical utility of the review. We are open to any other suggestions you might have.
Reviewer 3 Report
Comments and Suggestions for Authors
Regarding the manuscript titled Turning The Tide – Artificial Intelligence in the Evolving Landscape of Liver Cancer
I would like to inform the authors that I have reviewed this study and found that it lacks transparency, it is not clear whether it is a review or an original study. It is certainly not original but it does not meet the requirements of a review either. The final results of this study are not clear and the study methodology is not clear and the results are not clearly stated.
I strongly reject this study
Author Response
We thank the reviewer for this feedback and recognize that our initial submission may not have sufficiently conveyed the scope and nature of the manuscript. In the revised version, we have clearly indicated that this work is a narrative review synthesizing published evidence on artificial intelligence in liver cancer care, rather than an original study. To improve clarity, we introduced a dedicated Methods section, where we describe our search strategy, inclusion and exclusion criteria, and the process of categorizing studies by clinical task. The Results section has been organized to follow the clinical pathway of liver cancer management—early detection, diagnosis, staging, prognosis, treatment planning, and recurrence/surveillance—and within each subsection we summarize the evidence, report performance metrics where available, and provide comparative commentary against current standard-of-care approaches. We also added concluding synthesis statements to highlight the strongest performing AI/ML methods and validation strategies. Finally, the Conclusion has been expanded to clearly state the main findings. These revisions ensure that the manuscript now provides a transparent methodology, a clear synthesis of results, and a balanced discussion of outcomes, aligning with the standards of a review article.
We hope these revisions address the reviewer’s concerns regarding transparency, methodology, and clarity, and demonstrate that the manuscript now fulfills the requirements of a review article.
Reviewer 4 Report
Comments and Suggestions for Authors
The manuscript presents a comprehensive review of the role of artificial intelligence (AI) in liver cancer management, with a particular focus on treatment decision-making, recurrence prediction, and barriers to implementation. The authors synthesize findings from multiple studies applying machine learning (ML) and deep learning (DL) approaches particularly in hepatocellular carcinoma (HCC). The review addresses prognostic stratification, personalized therapeutic guidance, imaging-based prediction models, and post-treatment recurrence monitoring.
The manuscript is well-written. However, before acceptance, I would recommend some minor revisions.
- In the manuscript the authors addressed a critical point: the integration of AI into clinical workflow. In this section the authors should better comment on how AI may fit into real-world multidisciplinary tumor boards and decision-making workflows.
- Validation of diagnostic and prognostic models is another critical point. While prospective validation is emphasized in the manuscript, it would be interesting to specify how many of these published models have an external validation and how many models have a prospective validation.
Author Response
We thank the reviewer for the positive evaluation of our manuscript and for the constructive revision requests. We have revised the text accordingly, as detailed below.
Comment 1 - Response: We expanded the relevant section to include a discussion of AI integration into real-world clinical workflows.
Comment 2 - Response: We carefully reviewed all included studies and explicitly quantified validation types in the revised manuscript. The Results section now states:
- External validation: conducted in a limited number of studies, including Zhang et al. (JMRI, ref 59), Wang et al. (EJSO, ref 63), Xia et al. (Radiology, ref 64), Zhang et al. (Cancer Imaging, ref 67), Yamashita et al. (Sci Rep, ref 104), and Lai et al. (Cancer Commun, ref 111).
- Prospective validation: reported in only one multi-center study (Wei et al., Cancers, ref 61) and one single-center MRI radiomics study (Zhang et al., Cancer Imaging, ref 90).
We emphasize that while internal validation is common, external and prospective validations remain rare, underscoring the translational gap.
We thank the reviewer for these helpful comments, which we believe further strengthen the manuscript.
Round 2
Reviewer 1 Report
Comments and Suggestions for Authors
I am fine with this revision.
Reviewer 2 Report
Comments and Suggestions for Authors
The author address my concerns, so i accept the paper for publication.
Comments on the Quality of English LanguageNo comments
Reviewer 3 Report
Comments and Suggestions for Authors
All the revisions are done carefully. i can refers it for publication.